# Risk factors for abdominal aortic aneurysm in general populations: A systematic review and meta-analysis

**Pingzhen Zhang, Chunyan Liu, Jingjing Li, Huan Gao, Yuanyuan Fu, Wan Feng, Zhouzhen Chen, Xia Chen, Guiqin Wu**◉*

Department of Cardiovascular Surgery, Peking University Shenzhen Hospital, Shenzhen, China

* wgq0511@sina.cn

## Abstract

### Background

Abdominal aortic aneurysm (AAA) has a high mortality rate after rupture. This study systematically explored the risk factors associated with AAA in the general population using a meta-analytic approach.

### Materials and methods

We conducted a systematic search of PubMed, Embase, and Cochrane Library databases to identify relevant literature. The search was conducted through May 2025. Factors considered in more than three studies were included in the analysis. Odds ratios (ORs) with 95% confidence intervals (CIs) were used as effect estimates and all pooled analyses were performed using a random-effects model.

### Results

Thirty-four studies reporting 34,551 AAA cases were selected for meta-analysis. Increased risk of AAA was associated with male (OR: 3.78; 95% CI: 2.80–5.10; $P<0.001$), current or ever smoking (OR: 3.39; 95% CI: 2.57–4.48; $P<0.001$), hypertension (OR: 1.31; 95% CI: 1.21–1.42; $P<0.001$), dyslipidemia (OR: 1.33; 95% CI: 1.24–1.43; $P<0.001$), coronary artery disease (OR: 1.81; 95% CI: 1.66–1.98; $P<0.001$), cerebrovascular disease (OR: 1.32; 95% CI: 1.18–1.48; $P<0.001$), peripheral vascular disease (OR: 1.67; 95% CI: 1.47–1.91; $P<0.001$), chronic obstructive pulmonary disease (OR: 1.58; 95% CI: 1.31–1.90; $P<0.001$), renal disease (OR: 1.91; 95% CI: 1.28–2.83; $P=0.001$) and family history of AAA (OR: 2.26; 95% CI: 1.58–3.25; $P<0.001$). However, diabetes mellitus was associated with a reduced risk of AAA (OR: 0.84; 95% CI: 0.74–0.95; $P=0.007$). Furthermore, the risk of AAA was not affected by advanced age, alcohol intake, cancer, being overweight, or physical activity. The association between AAA risk and sex, smoking, hypertension, diabetes

**Data availability statement:** All relevant data are within the paper and its Supporting information files

**Funding:** The author(s) received no specific funding for this work.

**Competing interests:** The authors have declared that no competing interests exist.

mellitus, renal disease, and a family history of AAA differs between Eastern and Western countries.

## Conclusions

We systematically explored the risk factors for AAA. AAA represent a significant public health concern. Thus, early intervention and health education targeting these risk factors are necessary to prevent their occurrence.
**Registration:** INPLASY2023120024

## Background

Abdominal aortic aneurysm (AAA) refers to the permanent local dilation of the abdominal aorta, with a dilation diameter greater than approximately 3 cm or more than 50% of the normal diameter [1,2]. According to population-based studies, AAA affects 3.9%–7.2% of males and 1.0%–1.3% of females [3]. Although it is typically asymptomatic in the early stages, the mortality rate can reach 80% after rupture [4]. Considering that most AAAs are asymptomatic before rupture, disease screening is particularly important. Some countries have implemented AAA screening programs. However, while such programs have demonstrated effectiveness in reducing mortality and rupture rates by enabling timely diagnosis and reducing healthcare costs [5,6], exploring the risk factors for AAA can reduce its incidence by identifying high-risk populations and facilitating early interventions targeting modifiable risk factors.

The main risk factors for AAA include advanced age, male sex, smoking history, coronary heart disease, hypertension, peripheral artery disease, previous myocardial infarction, and family history of AAA [7]. In young patients, risk factors include genetics and inflammatory diseases, while risk factors for AAA in older individuals still need further investigation. Additionally, risk factors may differ owing to significant differences in the incidence of AAA among countries [8].

Several studies have identified potential predictors of AAA in the general population [9–11]. Kobeissi et al. [9] identified 20 studies and found that hypertension was associated with an increased risk of AAA; a 14% and 28% increase in the relative risk of AAA was observed for every 20 mm Hg and 10 mm Hg increase in systolic and diastolic blood pressure, respectively. Weng et al. [10] identified 17 studies and found that the A allele of interleukin-10 (−1082 G/A) was associated with an increased risk of AAA, while haptoglobin-1 could be regarded as a risk factor for AAA in the European population. Lampsas et al. [11] identified 11 studies and found that increased lipoprotein levels was associated with an increased risk of AAA independent of race. A comprehensive meta-analysis found that the risk factors for AAA include male sex, smoking, hypertension, coronary artery disease (CAD), and family history of AAA [12]. Previous studies focused primarily on a few known risk factors and failed to comprehensively cover all potential risk factors. Although Lampsas et al. [11] reported that the relationship between high lipoprotein levels and increased AAA risk is not affected by racial differences, the current study further explored whether there

are significant differences in risk factors among different populations, thereby providing a basis for prevention strategies in various regions. The most recent studies that met the inclusion criteria were included in the analysis. Therefore, the current systematic review and meta-analysis were performed to identify potential risk factors for AAA in the general population.

## Methods

### Search strategy and selection criteria

This study adhered to the guidelines outlined in the Preferred Reporting Items for Systematic Reviews and Meta-Analyses (PRISMA), which was issued in 2020 [13] (PRISMA Checklist), and was registered at INPLASY (No. INPLASY2023120024). We screened the PubMed, Embase, and Cochrane Library databases for eligible studies. The search was conducted through May 2025. The core search terms used were as follows: (1) "Abdominal Aortic Aneurysms" OR "Aneurysms, Abdominal Aortic" OR "Aortic Aneurysms, Abdominal" OR "Abdominal Aortic Aneurysm" OR "Aneurysm, Abdominal Aortic"; (2) "Screening" OR "Mass Screenings" OR "Screening, Mass" OR "Screenings, Mass" OR "Screenings"; (3) "Factor, Risk" OR "Factors, Risk" OR "Risk Factor" OR "Population at Risk" OR "Risk, Population at" OR "Populations at Risk" OR "Risk, Populations at" NOT "surgical repair" (S1 File). Studies reporting the risk factors for AAA were eligible for our study, and the publication language was restricted to English. Relevant review articles were manually searched for potential references to select studies that met the inclusion criteria.

The inclusion criteria for the study were as follows: (1) Participants: studies that examined the general adult population; (2) Exposure: studies that reported ≥3 risk factors for AAA; (3) Outcome: studies that reported effect estimates for risk factors related to AAA; and (4) Study design: cross-sectional, case-control, and cohort studies. When multiple studies of the same population were published, the most comprehensive study was selected based on several criteria. First, we compared the studies to confirm whether they involved the same or overlapping populations by examining study periods, locations, and participant characteristics. Next, we evaluated the comprehensiveness of each study by considering factors such as robustness of the study design, sample size, duration of follow-up, and completeness and quality of the data. The study with the largest sample size, longest follow-up period, and the most rigorous methodology was selected to ensure the inclusion of the most reliable and extensive data. Reviews and case report articles were removed because of insufficient data for quantitative analysis. Studies that primarily focused on surgical repair techniques without addressing risk factors or screening were excluded. Two authors (P Zhang and C Liu) independently conducted the literature search and screening process following a standardized protocol. In the case of any discrepancies between the two authors' results, the primary author (G Wu) was invited to make a final decision regarding study inclusion.

### Data collection and quality assessment

Two authors (J Li and H Gao) independently extracted the relevant information from the included studies according to standard procedures. The team discussed inconsistencies until consensus was reached. The following information was collected from each included study: first author's name, publication year, study design, study period, country, sample size, age, proportion of participants who were male, family history of AAA, number of AAAs, outcome investigated, and effect estimates for AAA risk factors (Data Extraction Table). When the data of the indicators required for calculation is missing, the partial information corresponding to that indicator is excluded from the analysis without affecting the use of other available information of the study.

The methodological rigor of the study was assessed using the Newcastle-Ottawa Scale (NOS), which evaluates three domains: selection (4 items), comparability (1 item), and outcome (3 items). The scale ranges from 0 to 9 [14]. Two reviewers independently assessed the quality of the individual trials. Any disagreements between the reviewers were resolved by the involvement of an additional reviewer.

## Statistical analysis

Before pooling the data, summary odds ratios (ORs) with 95% confidence intervals (CIs) were calculated for each study. Pooled analyses were conducted using a random-effects model, considering potential variability across the included studies [15,16]. Heterogeneity among the studies was evaluated using $I^2$ and Q statistics. When the $I^2$ value was ≥50% or the $P$-value was <0.10, we considered significant heterogeneity to be present [17,18]. Sensitivity analysis was performed to assess the stability of the pooled conclusions by sequentially removing individual studies [19]. Subgroup analyses were also performed according to country, and the differences between subgroups were compared using an interaction t-test, which assumed that the data distribution in the subgroups was normal [20]. Publication bias was assessed using funnel plots and Egger and Begg tests [21,22]. The reported $P$-values for the pooled conclusions were calculated using a two-sided test with a significance level of 0.05. Statistical analyses were conducted using STATA software (version 12.0; Stata Corporation, College Station, TX, USA).

## Results

### Literature search and study characteristics

A comprehensive literature search yielded 2,348 publications (Identified Studies), and 2,231 articles were excluded based on irrelevant titles or abstracts. A full-text evaluation was conducted on the remaining 117 studies, 83 of which were excluded based on the following criteria: insufficient data (n = 39), no appropriate controls (n = 29), and being a review (n = 15). Two potentially eligible studies were identified through a manual search of the relevant research references. However, upon review of the full texts, these two studies were found to have reported the same population. Thus, 34 studies met the inclusion criteria and were selected for the final analysis [23–56] (Fig 1).

The baseline characteristics of the identified studies and individuals involved are summarized in Table 1. Of the 34 included studies, seven had a prospective design, eight a retrospective design, and the remaining 19 had a cross-sectional design. Twenty-eight studies were conducted in Western countries, whereas the remaining six were conducted in Eastern countries. The included studies reported 34,551 cases of AAA, with sample sizes ranging from 421 to 3,056,455 individuals. Eight studies had an NOS score of eight, 14 had an NOS score of seven, and 12 had an NOS score of six (S1 Table).

### Association between risk factors and AAA risk

**Sex, smoking, and hypertension.** Associations between AAA risk and sex, smoking, and hypertension were reported in 16, 32, and 28 studies, respectively (Fig 2). Increased AAA risk as associated with being male (OR: 3.78; 95% CI: 2.80–5.10; $P$<0.001), current or ever smoking (OR: 3.39; 95% CI: 2.57–4.48; $P$<0.001), and hypertension (OR: 1.31; 95% CI: 1.21–1.42; $P$<0.001). There was significant heterogeneity among the included studies in terms of sex ($I^2$=89.2%, $P$<0.001), smoking status ($I^2$=97.7%, $P$<0.001), and hypertension ($I^2$=76.9%, $P$<0.001). Moreover, the sensitivity analysis indicated that the pooled conclusions regarding the association of sex, smoking, and hypertension with AAA risk were stable and remained unaltered after removing individual studies (S1–S3 Fig). Subgroup analyses revealed that the association between sex, smoking, and AAA risk was consistent with the overall analysis, and the risk in Western countries was greater than that in Eastern countries. No significant association was observed between hypertension and AAA risk in Eastern countries (Table 2). Significant publication bias was observed for sex, smoking, and hypertension (S4–S6 Fig).

**Diabetes mellitus, dyslipidemia, and coronary artery disease.** Associations between AAA risk and diabetes mellitus (DM), dyslipidemia, and coronary artery disease (CAD) were reported in 25, 17, and 24 studies, respectively (Fig 3). Increased risk of AAA was associated with dyslipidemia (OR: 1.33; 95% CI: 1.24–1.43; $P$<0.001) and CAD (OR: 1.81; 95% CI: 1.66–1.98; $P$<0.001), whereas DM played a protective role in AAA (OR: 0.84; 95% CI: 0.74–0.95; $P$=0.007).

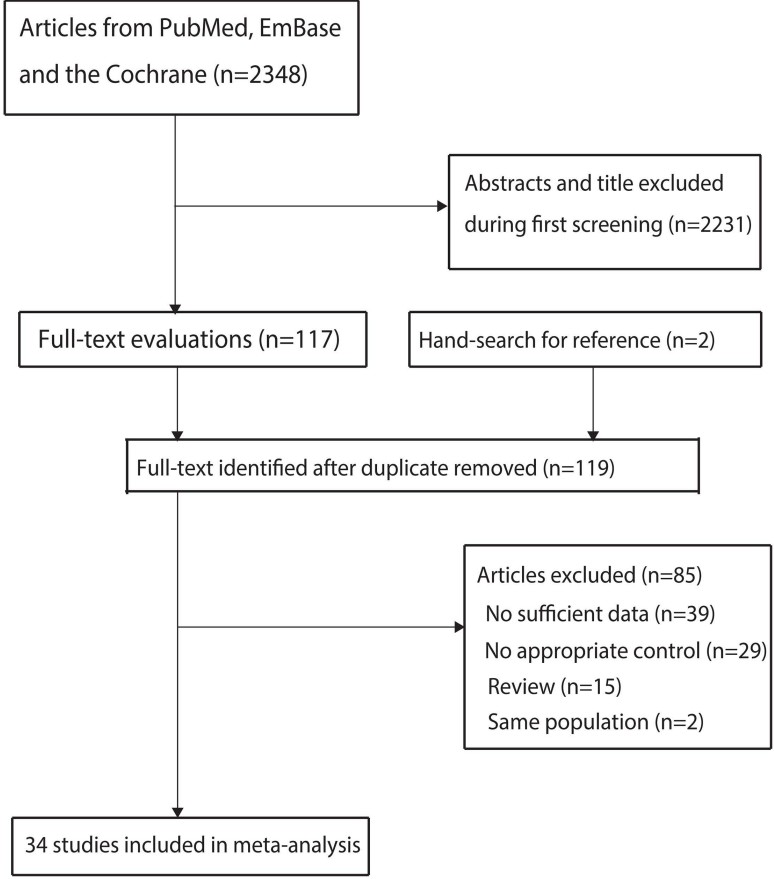

**Fig 1. PRISMAR flowchart for literature search and study selection.**

Significant heterogeneity was observed among the included studies for DM ($I^2$=74.6%, $P$<0.001), dyslipidemia ($I^2$=41.5%, $P$=0.038), and CAD ($I^2$=71.7%, $P$<0.001). The pooled conclusions regarding the association of DM, dyslipidemia, and CAD with AAA risk were robust (S7–S9 Fig). Subgroup analyses indicated a significant association between AAA risk and DM, dyslipidemia, and CAD, which was primarily observed in Western countries (Table 2). No significant publication bias was observed in the association between AAA risk and DM, dyslipidemia, or CAD (S10–S12 Fig).

**Cerebrovascular disease, peripheral vascular disease, and chronic obstructive pulmonary disease.** Associations between AAA risk and cerebrovascular disease, peripheral vascular disease (PVD), and chronic obstructive pulmonary disease (COPD) were reported in 13, 7, and 14 studies, respectively (Fig 4). Increased risk of AAA was associated with cerebrovascular disease (OR: 1.32; 95% CI: 1.18–1.48; $P$<0.001), PVD (OR: 1.67; 95% CI: 1.47–1.91; $P$<0.001), and COPD (OR: 1.58; 95% CI: 1.31–1.90; $P$<0.001), with significant heterogeneity observed for cerebrovascular disease ($I^2$=67.5%; $P$=0.016), and COPD ($I^2$=77.3%; $P$<0.001). The pooled conclusions regarding the associations of cerebrovascular disease, PVD, and COPD with AAA risk were stable (S13–S15 Fig). Subgroup analyses found that cerebrovascular disease and PVD were risk factors for AAA in Western countries, whereas COPD was identified as a risk factor in both Eastern and Western countries (Table 2). Potentially significant publication bias was observed only for COPD, whereas no bias was found for cerebrovascular disease or PVD (S16–S18 Fig).

**Table 1. The baseline characteristics of included studies and participants involved.**

| Study | Study design | Study period | Country | Sample size | Age (years) | Male (%) | Family history of AAA | No of AAA | Outcome investigated | Study quality |
|---|---|---|---|---|---|---|---|---|---|---|
| Smith 1993 [23] | Cross-sectional | 1989 | UK | 2,597 | 65-74 | NA | NA | 219 | CAD, PVD, COPD, DM, hypertension, smoking, dyslipidemia | 7 |
| Simoni 1995 [24] | Cross-sectional | 1991-1994 | Italy | 1,601 | 65-75 | 46.3 | NA | 70 | CAD, COPD, DM, hypertension, smoking, dyslipidemia, alcohol intake | 6 |
| Lederle 1997 [25] | Cross-sectional | 1992-1995 | USA | 73,451 | 50-79 | 97.2 | 5.1 | 3,366 | CAD, COPD, DM, hypertension, smoking, dyslipidemia, sex, family history of AAA, cerebrovascular disease, cancer | 8 |
| Vazquez 1998 [26] | Cross-sectional | 1995-1996 | Belgium | 727 | 65-75 | 100.0 | NA | 33 | DM, hypertension, smoking, dyslipidemia | 6 |
| Lederle 2000 [27] | Cross-sectional | 1995-1997 | USA | 52,745 | 50-79 | 97.4 | 5.0 | 1,917 | CAD, COPD, DM, hypertension, smoking, dyslipidemia, sex, family history of AAA, cerebrovascular disease, cancer | 8 |
| Singh 2001 [28] | Cross-sectional | 1994-1995 | Norway | 6,386 | 25-84 | 46.4 | NA | 337 | hypertension, smoking, age, physical activity | 8 |
| Kent 2010 [29] | Retrospective | 2003-2008 | USA | 3,056,455 | < 85 | 35.3 | 2.5 | 23,446 | CAD, PVD, DM, hypertension, dyslipidemia, sex, family history of AAA, cerebrovascular disease, age | 7 |
| Svensjo 2011 [30] | Cross-sectional | 2009 | Sweden | 14,611 | 65 | 100.0 | NA | 233 | CAD, COPD, hypertension, smoking, dyslipidemia, cerebrovascular disease | 7 |
| Barba 2013 [31] | Cross-sectional | 2008-2010 | Spain | 781 | 65 | 100.0 | NA | 37 | PVD, DM, hypertension, smoking, family history of AAA, cerebrovascular disease, renal disease | 6 |
| Hager 2013 [32] | Cross-sectional | 2008-2010 | Sweden | 4,715 | 70 | 100.0 | NA | 107 | CAD, COPD, smoking, dyslipidemia, renal disease, cancer | 7 |
| Svensjo 2013 [33] | Cross-sectional | 2007-2009 | Sweden | 5139 | 70 | 0.0 | 2.4 | 19 | CAD, COPD, DM, hypertension, smoking, dyslipidemia, family history of AAA, cerebrovascular disease, renal disease | 7 |
| Jawien 2014 [34] | Cross-sectional | 2009-2011 | Poland | 1,556 | 60-92 | 100.0 | NA | 94 | smoking, family history of AAA | 6 |
| Chun 2014 [35] | Retrospective | 2007-2009 | USA | 6,142 | 65-75 | 99.6 | NA | 496 | CAD, PVD, COPD, DM, hypertension, smoking, dyslipidemia, renal disease, age, overweight | 7 |
| Bohlin 2014 [36] | Prospective | 2006-2011 | Sweden | 1,443 | 65 | 100.0 | NA | 292 | CAD, COPD, DM, hypertension | 7 |
| Golledge 2014 [37] | Cross-sectional | 1996-1999 | Australia | 11,742 | 65-79 | 100.0 | NA | 931 | CAD, DM, hypertension, smoking, dyslipidemia, cerebrovascular disease, physical activity | 7 |
| Jahangir 2015 [38] | Prospective | 1999-2012 | USA | 18,501 | > 65 | 36.1 | NA | 281 | CAD, DM, hypertension, smoking, dyslipidemia, sex | 8 |
| Salvador-Gonzalez 2016 [39] | Cross-sectional | 2007-2010 | Spain | 651 | 65-74 | 100.0 | NA | 15 | CAD, smoking | 6 |
| Corrado 2016 [40] | Prospective | 2010-2013 | Italy | 1,555 | 60-85 | 48.6 | 6.7 | 22 | CAD, PVD, DM, hypertension, smoking, dyslipidemia, sex, family history of AAA | 7 |
| Kvist 2017 [41] | Cross-sectional | 2014-2015 | Denmark | 1,318 | 65-74 | 51.4 | NA | 91 | DM, hypertension, smoking, dyslipidemia, sex | 7 |
| Siso-Almirall 2017 [42] | Prospective | 2013-2014 | Spain | 1,010 | > 60 | 100.0 | 1.3 | 11 | CAD, COPD, DM, hypertension, smoking, dyslipidemia, family history of AAA, cerebrovascular disease, renal disease, overweight, physical activity | 8 |

*(Continued)*

**Table 1.** (Continued)

| Study | Study design | Study period | Country | Sample size | Age (years) | Male (%) | Family history of AAA | No of AAA | Outcome investigated | Study quality |
|---|---|---|---|---|---|---|---|---|---|---|
| Stackelberg 2017 [43] | Prospective | 2006-2011 | Sweden | 14,249 | 65-75 | 100.0 | NA | 168 | CAD, DM, hypertension, smoking, dyslipidemia, overweight, alcohol intake | 8 |
| Han 2017 [44] | Cross-sectional | NA | Korea | 2,035 | 23-95 | 44.6 | NA | 22 | COPD, smoking, sex, alcohol intake | 6 |
| Li 2018 [45] | Cross-sectional | 2014-2015 | China | 5,402 | > 40 | 47.3 | 0.1 | 18 | CAD, DM, hypertension, smoking, family history of AAA, cerebrovascular disease, alcohol intake | 7 |
| Kilic 2018 [46] | Retrospective | 2016-2017 | Turkey | 1,876 | > 65 | 50.1 | 0.9 | 69 | CAD, COPD, smoking, sex, family history of AAA | 6 |
| Song 2020 [47] | Retrospective | 1997-2012 | Korea | 1,658 | 61 | 86.4 | NA | 271 | COPD, DM, hypertension, smoking, sex, renal disease | 6 |
| Vats 2020 [48] | Retrospective | 2010-2017 | Sweden | 421 | 65 | 100.0 | NA | 142 | CAD, DM, hypertension, smoking, dyslipidemia, cancer | 6 |
| Summers 2021 [49] | Retrospective | 2001-2017 | USA | 9,457 | > 45 | 47.4 | 22.4 | 267 | CAD, DM, hypertension, smoking, dyslipidemia, sex, family history of AAA, age, overweight | 7 |
| Wiles 2021 [50] | Retrospective | 2002-2017 | USA | 814 | 34-97 | 47.0 | NA | 90 | hypertension, smoking, sex | 6 |
| Obel 2021 [51] | Prospective | 2014-2018 | Denmark | 14,989 | 60-74 | 95.0 | NA | 540 | CAD, PVD, DM, hypertension, smoking, sex, cerebrovascular disease, renal disease | 8 |
| Kim 2023 [52] | Cross-sectional | 2008-2019 | Korea | 3,124 | 61-75 | 43.4 | NA | 22 | smoking, sex, cerebrovascular disease, renal disease | 6 |
| Lin 2023 [53] | Cross-sectional | 2019-2021 | China | 9,559 | 70.3 | 64.3 | NA | 219 | CAD, PVD, DM, hypertension, smoking, sex, cerebrovascular disease | 7 |
| Koncar 2024 [54] | Cross-sectional | 2023 | Serbia | 4,046 | 68.8 | 51.2 | 9.6 | 195 | Sex, family history of AAA, hypertension, DM, smoking, overweight, alcohol intake | 6 |
| Stacey 2024 [55] | Retrospective | 1993-2015 | UK | 6,879 | 60-80 | 54.2 | 4.0 | 275 | Sex, hypertension, smoking, CAD, cerebrovascular disease, COPD, DM, family history of AAA | 7 |
| Persson 2025 [56] | Prospective | 1986-2010 | Sweden | 1,171 | 57.0 | 76.7 | NA | 236 | Overweight, hypertension, DM, CAD, smoking | 8 |

**Renal disease and family history of AAA.** An association between AAA risk and renal disease, as well as a family history of AAA, was reported in eight and 13 studies, respectively (Fig 5). The summary results indicated that increased risk of AAA was associated with renal disease (OR: 1.91; 95% CI: 1.28–2.83; $P=0.001$) and family history of AAA (OR: 2.26; 95% CI: 1.58–3.25; $P<0.001$). There was significant heterogeneity among the included studies for renal disease ($I^2=70.4\%$; $P=0.001$) and family history of AAA ($I^2=97.1\%$; $P<0.001$). The pooled conclusions for the association of renal disease and family history of AAA with the risk of AAA were stable (S19–S20 Fig). The strength of the association between renal disease and AAA risk was greater in Eastern countries than in Western countries, whereas family history was considered a risk factor in Western countries (Table 2). No significant publication bias was observed for renal disease or family history of AAA (S21–S22 Fig).

**Age, alcohol intake, cancer, overweightness, and physical activity.** An association between AAA risk and advanced age, alcohol intake, cancer, overweight, and physical activity was reported in 4, 5,4, 6, and 3 studies, respectively (Fig 6). There was no significant association between AAA risk and advanced age (OR: 4.17; 95% CI: 0.97–17.99; $P=0.055$), alcohol intake (OR: 1.09; 95% CI: 0.78–1.52; $P=0.621$), cancer (OR: 0.90; 95% CI: 0.73–1.11;

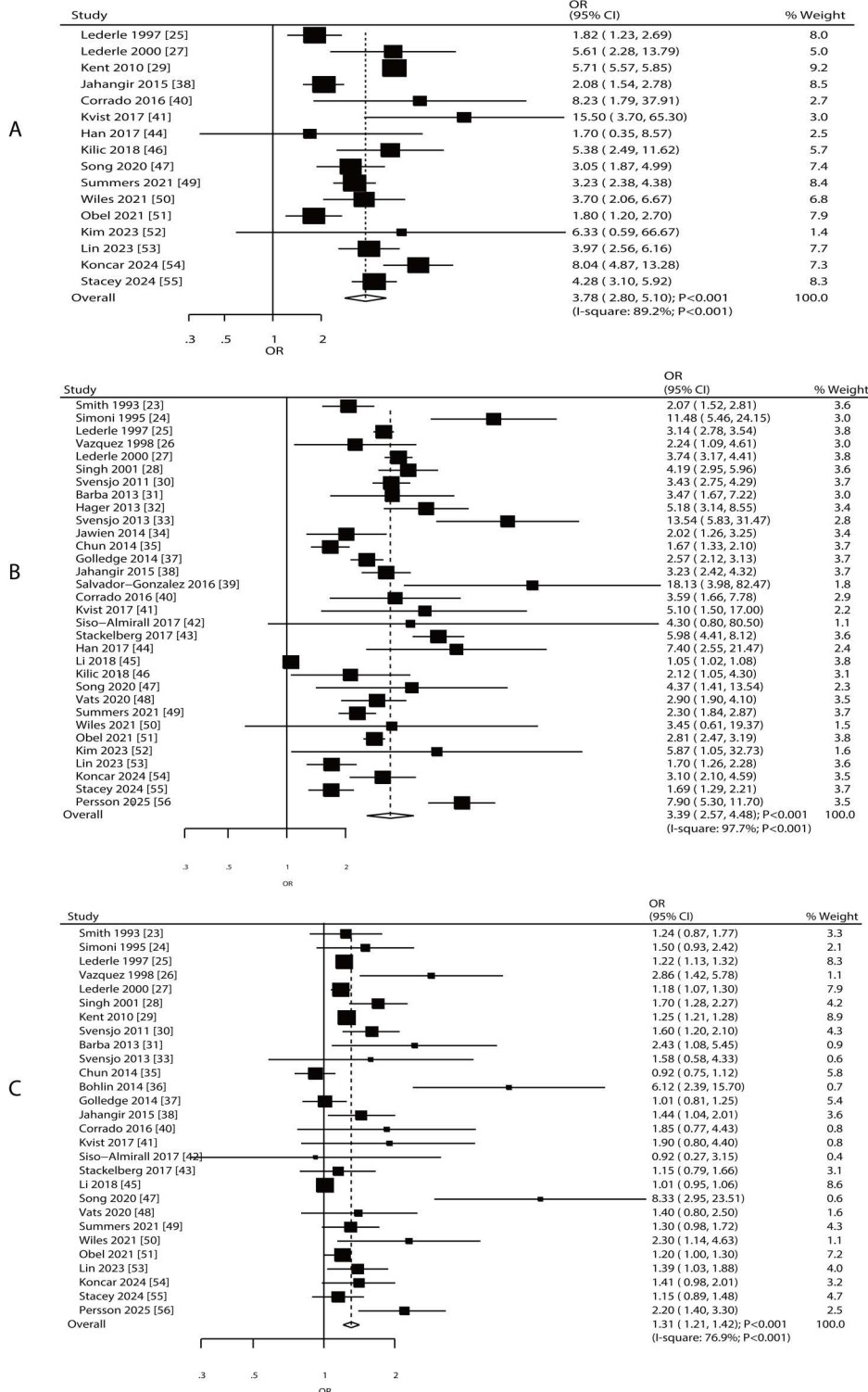

**Fig 2. Association between AAA risk and sex, smoking, and hypertension. A: sex; B: smoking; C: hypertension.**

**Table 2. Subgroup analyses for risk factors according to country.**

| Factors | Subgroup | No of studies | OR and 95%CI | P value | $I^2$ (%) | P value for $I^2$ | P value between subgroups |
|---|---|---|---|---|---|---|---|
| Sex (male vs female) | Eastern | 3 | 3.57 (2.58-4.94) | < 0.001 | 0.0 | 0.656 | 0.007 |
| | Western | 13 | 3.82 (2.71-5.38) | < 0.001 | 90.8 | < 0.001 | |
| Current or ever smoking | Eastern | 6 | 2.28 (1.35-3.84) | 0.002 | 86.3 | < 0.001 | < 0.001 |
| | Western | 26 | 3.35 (2.86-3.92) | < 0.001 | 83.5 | < 0.001 | |
| Hypertension | Eastern | 3 | 1.67 (0.94-2.98) | 0.080 | 90.0 | < 0.001 | < 0.001 |
| | Western | 25 | 1.29 (1.20-1.39) | < 0.001 | 56.6 | < 0.001 | |
| DM | Eastern | 3 | 1.04 (0.72-1.50) | 0.838 | 35.8 | 0.210 | 0.027 |
| | Western | 22 | 0.81 (0.71-0.93) | 0.003 | 75.7 | < 0.001 | |
| Dyslipidemia | Eastern | 0 | – | – | – | – | – |
| | Western | 17 | 1.33 (1.24-1.43) | < 0.001 | 41.5 | 0.038 | |
| CAD | Eastern | 3 | 1.20 (0.53-2.74) | 0.664 | 86.3 | 0.001 | 0.609 |
| | Western | 21 | 1.81 (1.66-1.98) | < 0.001 | 69.9 | < 0.001 | |
| Cerebrovascular disease | Eastern | 3 | 1.05 (0.31-3.54) | 0.933 | 84.9 | 0.001 | 0.740 |
| | Western | 10 | 1.31 (1.19-1.45) | < 0.001 | 61.9 | 0.005 | |
| PVD | Eastern | 1 | 1.10 (0.54-2.24) | 0.793 | – | – | 0.305 |
| | Western | 5 | 1.70 (1.48-1.94) | < 0.001 | 46.6 | 0.095 | |
| COPD | Eastern | 3 | 1.53 (1.11-2.11) | 0.010 | 17.0 | 0.300 | 0.190 |
| | Western | 11 | 1.58 (1.27-1.96) | < 0.001 | 81.2 | < 0.001 | |
| Renal disease | Eastern | 2 | 3.55 (1.61-7.82) | 0.002 | 8.3 | 0.296 | 0.019 |
| | Western | 6 | 1.61 (1.08-2.41) | 0.019 | 70.7 | 0.004 | |
| Family history of AAA | Eastern | 2 | 1.49 (0.91-2.44) | 0.112 | 43.6 | 0.183 | < 0.001 |
| | Western | 11 | 2.43 (1.74-3.39) | < 0.001 | 94.6 | < 0.001 | |

$P = 0.332$), overweight (OR: 1.23; 95% CI: 0.93–1.62; $P = 0.146$), and physical activity (OR: 0.91; 95% CI: 0.83–1.00; $P = 0.054$). Significant heterogeneity was observed for advanced age ($I^2 = 99.6\%$; $P < 0.001$), alcohol intake ($I^2 = 73.4\%$; $P = 0.005$), cancer ($I^2 = 65.0\%$; $P = 0.036$), and overweight ($I^2 = 70.8\%$; $P = 0.004$); however, no evidence of heterogeneity for physical activity was observed ($I^2 = 0.0\%$; $P = 0.434$).

## Discussion

This study systematically explored the risk factors for AAA. Thirty-four studies worldwide reported 34,551 AAA cases that met the inclusion criteria. Despite the wide variations in the characteristics of the included populations, a meta-analysis based on several studies enabled stable conclusions to be drawn.

This study found that risk factors for AAA were male sex, current or ever smoking, hypertension, dyslipidemia, CAD, cerebrovascular disease, PVD, COPD, renal disease, and a family history of AAA. Guirguis-Blake et al. [3] indicated that AAA is more prevalent in males than in females, which could be explained by several factors, including differences in hormonal profiles, smoking prevalence, and other lifestyle factors [57,58]. Further research is required to fully understand the underlying mechanisms. Wilson et al. [59] found that nicotine may activate tissue plasminogen activators, inducing macrophages to produce matrix metalloproteinases and disrupt collagen synthesis, whereas Kugo et al. [60] noted that nicotine can weaken the vascular wall, increase collagenolytic activity, and promote the degradation of abdominal aortic elastin and collagen. Hypertension and dyslipidemia were identified as risk factors. Rapsomaniki et al. [61] reported that hypertension is associated with arterial stiffness, which could affect the risk of AAA. Additionally, along with DM and CAD, hypertension is a common risk factor for COPD and AAA [62]. In contrast, Chen et al. [63] noted that dyslipidemia is significantly related to AAA activity via induced endothelial dysfunction. Along with smoking and atherosclerosis, dyslipidemia

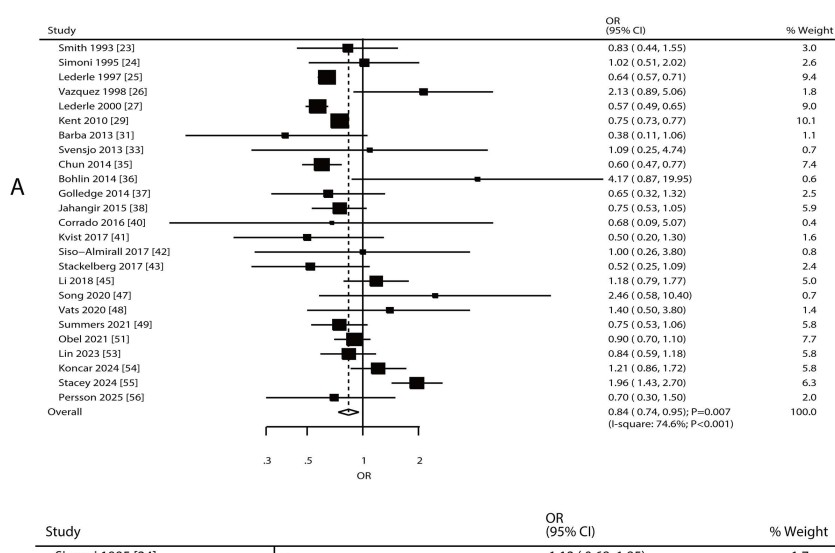

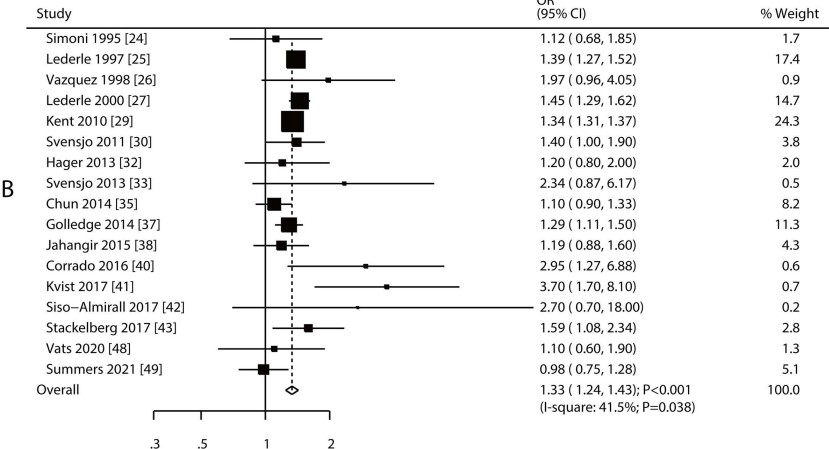

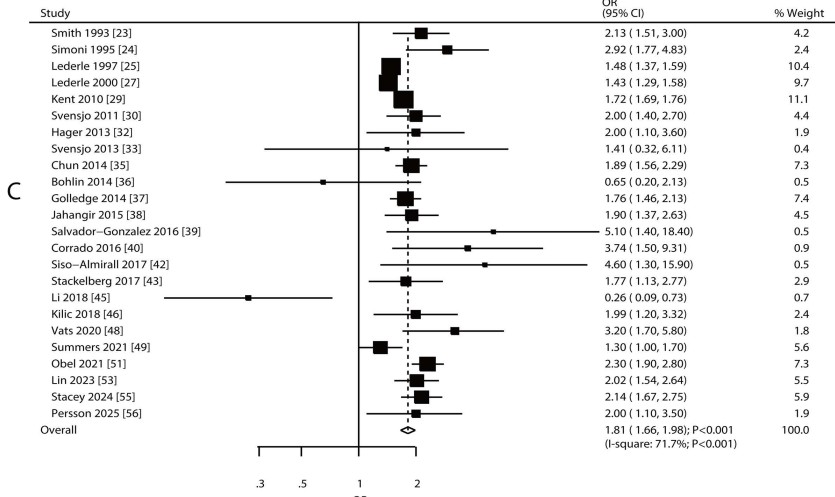

**Fig 3. Association between AAA risk and DM, dyslipidemia, and CAD. A: DM; B: dyslipidemia; C: CAD.**

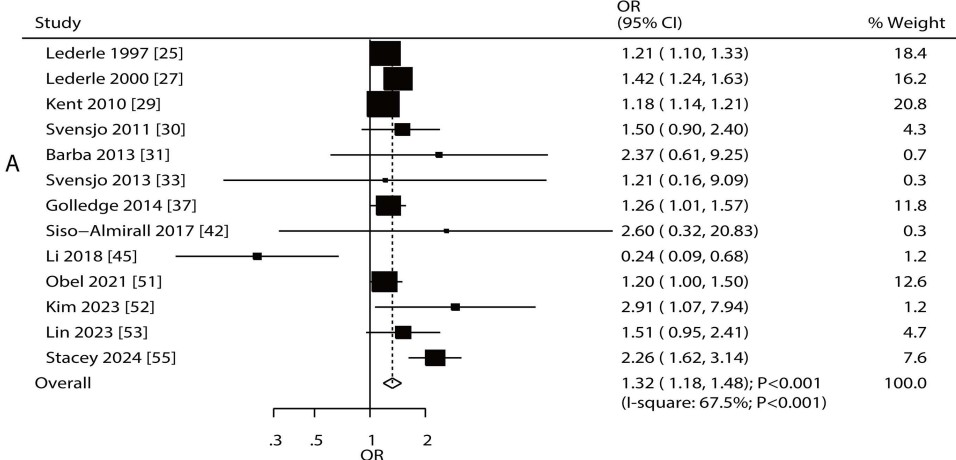

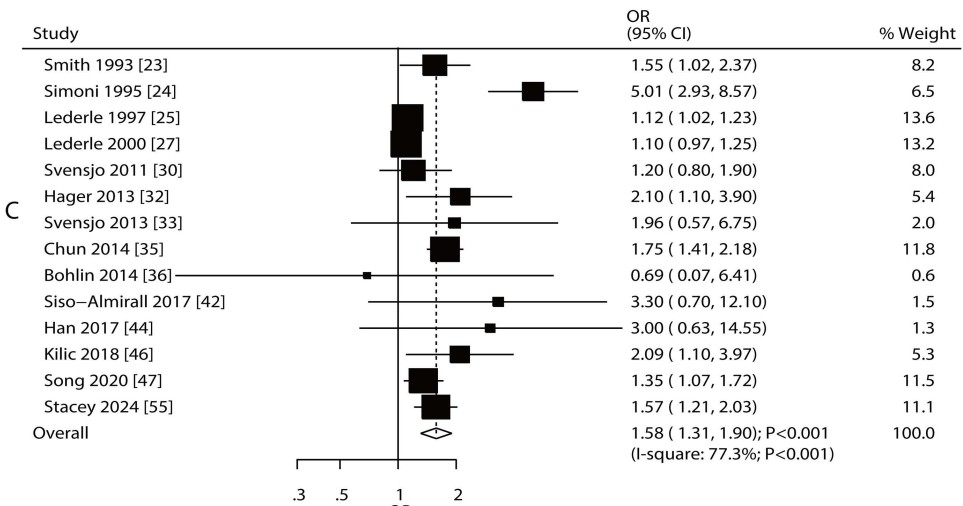

**Fig 4. Association between AAA risk and cerebrovascular disease, PVD, and COPD. A: cerebrovascular disease; B: PVD; C: COPD.**

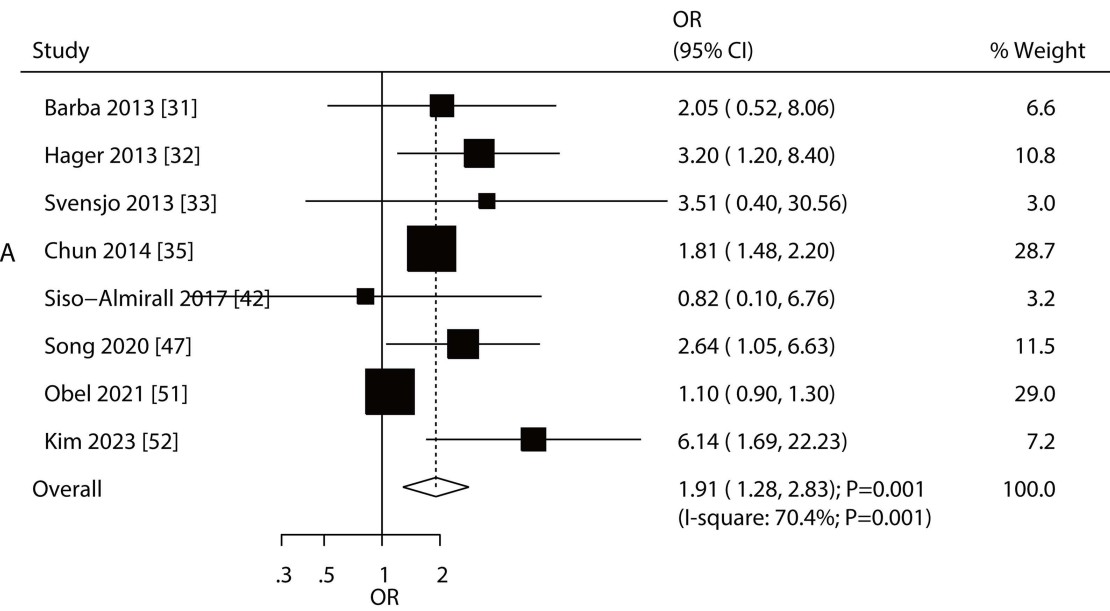

**Fig 5. Association between AAA risk and renal disease and family history of AAA. A: renal disease; B: family history of AAA.**

is also a common risk factor for CAD, cerebrovascular disease, PVD, and AAA [64]. In renal disease, infected patients are more prone to developing abnormal neovascularization in the aorta, which plays a crucial role in the pathophysiology of AAA. In addition, renal disease causes systemic inflammation, which is believed to increase the risk of AAA [65]. Finally, the increased risk of AAA in patients with a family history of AAA may be related to inflammatory markers and impaired endothelial function as well as genetic susceptibility and environmental risk factors [66]. These findings highlight the

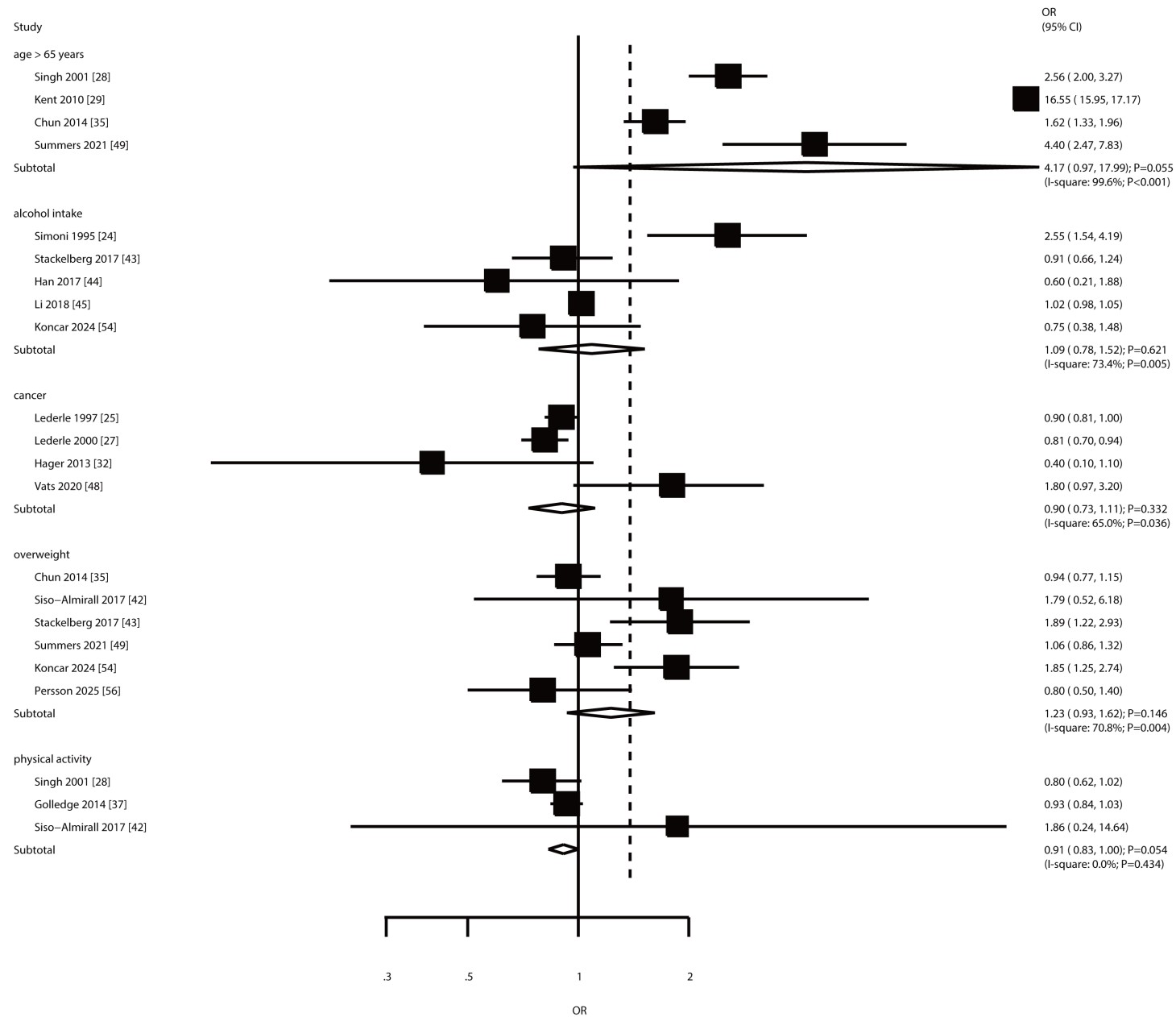

**Fig 6. Association between AAA risk and age, alcohol intake, cancer, overweight, and physical activity.**

importance of early intervention and health education for improving modifiable risk factors to reduce the risk of AAA, primarily including: (1) implementing targeted screening programs for high-risk populations, particularly males over 65 years of age, current or former smokers, and individuals with a family history of AAA; (2) establishing regular follow-up protocols to monitor the progression and provide early intervention if necessary for patients identified with small AAAs; (3) risk factor management, including smoking cessation, and hypertension and dyslipidemia management; and (4) health education regarding lifestyle modifications and campaigns to promote awareness of signs and symptoms of AAA.

Different risk factors for AAA were found between Eastern and Western countries, including sex, smoking, hypertension, DM, renal disease, and family history. This could be explained by the following: (1) the prevalence of AAA is higher in families, with genetic factors playing a significant role in its occurrence. Variations in genetic backgrounds among different ethnicities may lead to differences in disease prevalence across regions; (2) dietary structure, exercise habits, smoking, and other lifestyle factors affect the occurrence of AAA. There are differences in lifestyle between Western and Eastern countries, with higher smoking rates in Western countries and the dietary structure in Eastern countries potentially being related to the occurrence of AAA; (3) the characteristics of disease prevalence may vary among populations in different regions; for instance, the proportions of individuals with conditions such as hypertension and hyperlipidemia differ between Eastern and Western countries, and these conditions are somewhat associated with the occurrence of AAA; (4) the different levels of screening and treatment for AAA in different countries may affect the incidence and mortality rates; and (5) only five of the included studies were performed in Eastern countries, and the pooled conclusions were not stable.

In contrast to the risk factors identified, this study found that AAA risk was not affected by advanced age, alcohol intake, cancer, being overweight, or physical activity, which could be explained by the small number of included studies, and the pooled conclusions were not variable. Furthermore, DM is associated with a reduced risk of AAA. Aortic wall stiffness increases and aortic wall remodeling decreases in patients with DM, while matrix loss in the small aortic wall of the aneurysm increases. Therefore, other explanations for the negative correlation between AAA growth and prevalence may be related to the medication treatment of patients with DM [67,68].

This study has several limitations that should be considered when interpreting the findings. First, we included a combination of cross-sectional, retrospective, and prospective studies. This heterogeneity in study design may have introduced selection bias, as different study types have varying levels of methodological rigor and are subject to different types of bias. Moreover, retrospective studies, in particular, are susceptible to recall bias, whereby participants may not accurately remember past exposures or events, potentially leading to misclassification of risk factors. Furthermore, the inclusion of observational studies means that uncontrolled confounding factors may have influenced the results. These confounders could not be fully accounted for in the meta-analysis, potentially leading to biased estimates of the association between risk factors and AAA. Second, most of the included studies followed a cross-sectional design, which limited our ability to establish causality. Moreover, the methodological quality of the included studies varied. Some studies had small sample sizes, limited follow-up periods, and inadequate control of confounding variables, which may have affected the reliability and generalizability of the findings. Third, there was substantial heterogeneity in most of the identified risk factors, which could not be fully explained by sensitivity and subgroup analyses. Finally, as with any meta-analysis, there is the potential for publication bias, whereby studies with significant or positive results are more likely to be published. Moreover, the meta-analyses were based on published articles, which restricted detailed analyses. Further large-scale prospective studies are required to verify the findings of this study and address the limitations mentioned above. Prospective studies with standardized data collection methods and rigorous control of confounding factors would help establish causality and provide more reliable estimates of risk factors for AAA.

## Conclusions

This study found that AAA risk was elevated in patients with the following characteristics: male sex, current or ever smoking, hypertension, dyslipidemia, CAD, cerebrovascular disease, PVD, COPD, renal disease, and family history of AAA. DM can be considered a protective factor against AAA. Furthermore, the roles of sex, smoking, hypertension, DM, renal disease, and family history in the risk of AAA may differ between Eastern and Western countries.

## Supporting information

**S1 File. Search strategies in PubMed, EmBase, and the Cochrane library.**
(DOCX)

**S2 File. PRISMA_2020_checklist.**
(DOCX)

**S1 Data. Data Extraction Table.**
(XLSX)

**S2 Data. Identified Studies.**
(XLSX)

**S1 Table. Quality scores of prospective cohort studies using Newcastle-Ottawa Scale.**
(DOCX)

**S1 Fig. Sensitivity analysis for male vs female on the risk of AAA.**
(JPG)

**S2 Fig. Sensitivity analysis for current or ever smoking with the risk of AAA.**
(JPG)

**S3 Fig. Sensitivity analysis for hypertension with the risk of AAA.**
(JPG)

**S4 Fig. Funnel plot for male vs female on the risk of AAA.**
(JPG)

**S5 Fig. Funnel plot for current or ever smoking with the risk of AAA.**
(JPG)

**S6 Fig. Funnel plot for hypertension with the risk of AAA.**
(JPG)

**S7 Fig. Sensitivity analysis for DM with the risk of AAA.**
(JPG)

**S8 Fig. Sensitivity analysis for dyslipidemia with the risk of AAA.**
(JPG)

**S9 Fig. Sensitivity analysis for CAD with the risk of AAA.**
(JPG)

**S10 Fig. Funnel plot for DM with the risk of AAA.**
(JPG)

**S11 Fig. Funnel plot for dyslipidemia with the risk of AAA.**
(JPG)

**S12 Fig. Funnel plot for CAD with the risk of AAA.**
(JPG)

**S13 Fig. Sensitivity analysis for cerebrovascular disease with the risk of AAA.**
(JPG)

**S14 Fig. Sensitivity analysis for PVD with the risk of AAA.**
(JPG)

**S15 Fig. Sensitivity analysis for COPD with the risk of AAA.**
(JPG)

**S16 Fig. Funnel plot for cerebrovascular disease with the risk of AAA.**
(JPG)

**S17 Fig. Funnel plot for PVD with the risk of AAA.**
(JPG)

**S18 Fig. Funnel plot for COPD with the risk of AAA.**
(JPG)

**S19 Fig. Sensitivity analysis for renal disease with the risk of AAA.**
(JPG)

**S20 Fig. Sensitivity analysis for family history of AAA with the risk of AAA.**
(JPG)

**S21 Fig. Funnel plot for renal disease with the risk of AAA.**
(JPG)

**S22 Fig. Funnel plot for family history of AAA with the risk of AAA.**
(JPG)

## Author contributions

**Conceptualization:** Pingzhen Zhang.

**Data curation:** Jingjing Li, Huan Gao, Yuanyuan Fu, Wan Feng, Zhouzhen Chen, Xia Chen.

**Formal analysis:** Chunyan Liu.

**Visualization:** Pingzhen Zhang.

**Writing – original draft:** Guiqin Wu.

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
