## [Decision Letter · Decision Letter 0]

15 Nov 2024

Dear Dr. Wu,

Thank you for submitting your manuscript to PLOS ONE. After careful consideration, we feel that it has merit but does not fully meet PLOS ONE’s publication criteria as it currently stands. Therefore, we invite you to submit a revised version of the manuscript that addresses the points raised during the review process.

We look forward to receiving your revised manuscript.

Kind regards,

Irena Ilic, MD, PhD

Academic Editor

PLOS ONE

Journal Requirements:

2. We note that your Data Availability Statement is currently as follows: “All relevant data are within the manuscript and in Supporting Information files.”

Please confirm at this time whether or not your submission contains all raw data required to replicate the results of your study. Authors must share the “minimal data set” for their submission. PLOS defines the minimal data set to consist of the data required to replicate all study findings reported in the article, as well as related metadata and methods (https://journals.plos.org/plosone/s/data-availability#loc-minimal-data-set-definition). For example, authors should submit the following data: - The values behind the means, standard deviations and other measures reported; - The values used to build graphs; - The points extracted from images for analysis. Authors do not need to submit their entire data set if only a portion of the data was used in the reported study. If your submission does not contain these data, please either upload them as Supporting Information files or deposit them to a stable, public repository and provide us with the relevant URLs, DOIs, or accession numbers. For a list of recommended repositories, please see https://journals.plos.org/plosone/s/recommended-repositories. If there are ethical or legal restrictions on sharing a de-identified data set, please explain them in detail (e.g., data contain potentially sensitive information, data are owned by a third-party organization, etc.) and who has imposed them (e.g., an ethics committee). Please also provide contact information for a data access committee, ethics committee, or other institutional body to which data requests may be sent. If data are owned by a third party, please indicate how others may request data access.

4. As required by our policy on Data Availability, please ensure your manuscript or supplementary information includes the following: A numbered table of all studies identified in the literature search, including those that were excluded from the analyses. For every excluded study, the table should list the reason(s) for exclusion. If any of the included studies are unpublished, include a link (URL) to the primary source or detailed information about how the content can be accessed. A table of all data extracted from the primary research sources for the systematic review and/or meta-analysis. The table must include the following information for each study: Name of data extractors and date of data extraction Confirmation that the study was eligible to be included in the review. All data extracted from each study for the reported systematic review and/or meta-analysis that would be needed to replicate your analyses. If data or supporting information were obtained from another source (e.g. correspondence with the author of the original research article), please provide the source of data and dates on which the data/information were obtained by your research group. If applicable for your analysis, a table showing the completed risk of bias and quality/certainty assessments for each study or outcome. Please ensure this is provided for each domain or parameter assessed. For example, if you used the Cochrane risk-of-bias tool for randomized trials, provide answers to each of the signalling questions for each study. If you used GRADE to assess certainty of evidence, provide judgements about each of the quality of evidence factor. This should be provided for each outcome. An explanation of how missing data were handled. This information can be included in the main text, supplementary information, or relevant data repository. Please note that providing these underlying data is a requirement for publication in this journal, and if these data are not provided your manuscript might be rejected.

Reviewers' comments:

Reviewer's Responses to Questions

**Comments to the Author**

1. Is the manuscript technically sound, and do the data support the conclusions?

Reviewer #1: Yes

Reviewer #2: Partly

2. Has the statistical analysis been performed appropriately and rigorously?

Reviewer #1: Yes

Reviewer #2: Yes

3. Have the authors made all data underlying the findings in their manuscript fully available?

Reviewer #1: Yes

Reviewer #2: No

4. Is the manuscript presented in an intelligible fashion and written in standard English?

Reviewer #1: Yes

Reviewer #2: Yes

Reviewer #1: Major Revision Comments

1. Introduction:

o Contextualization of Previous Studies: The introduction provides a general overview of AAA and its risk factors. However, the manuscript would benefit from a more detailed discussion of the gaps in previous studies and the unique contributions of the current study. Please elaborate on the limitations of prior research and explicitly state how this study addresses these gaps.

2. Methods:

o Search Strategy: While the search strategy is described, the manuscript should include the exact search strings used for each database. This transparency allows for reproducibility of the search process.

o Study Selection Criteria: The inclusion and exclusion criteria should be more detailed. For example, clarify whether studies without full-text availability were included or excluded, and explain how studies with overlapping populations were handled beyond selecting the most comprehensive one.

3. Results:

o Study Characteristics: Provide a more comprehensive table summarizing the characteristics of the included studies (e.g., Outcome investigated, study duration key findings etc.).

o Subgroup Analyses: The manuscript mentions differences in risk factors between Eastern and Western countries but does not provide detailed subgroup analysis results. Include detailed subgroup analysis results and discuss any significant variations found.

4. Discussion:

o Implications for Practice and Policy: The manuscript concludes that early intervention and health education are necessary but lacks specific recommendations. Provide more detailed suggestions for clinical practice and public health policies based on the study’s findings.

o Limitations: The limitations section should be expanded. Discuss potential biases in study selection, limitations in the quality of included studies, and the potential impact of these limitations on the findings.

Reviewer #2: In this manuscript, the authors conducted a systematic review and meta-analysis on AAA, which is a common topic in vascular surgery. However, there are many similar systematic reviews and meta-analysis on AAA have already been published, and the majority of risk factors presented in this manuscript have been examined by other analysis, undermining the importance of the current study.

Major comments:

1, Novelty. What is the novelty of this current research? Most of these have been reported by other papers.

2, Because most of these have been reported repeatedly, I suggest the authors focus on the limitations of other meta-analysis while doing their own analysis.

3, There are multiple statements in the manuscript seeming to be more arbitrary than fact based. For example, line237-239:"AAA was more prevalent in males than in female, which

could explained by the arterial wall in males is thinner and weaker, making it more susceptible to

conditions like atherosclerosis, thus increasing the risk of AAA." what is the evidence for the claim that the arterial wall in males is thinner and weaker?

4. Multiple grammar mistakes need to be corrected.

**Do you want your identity to be public for this peer review?** For information about this choice, including consent withdrawal, please see our Privacy Policy

Reviewer #1: **Yes: ** Pradeep Kumar

Reviewer #2: No

---

## [Author Response · Author response to Decision Letter 1]

12 Jun 2025

PLOS ONE

RE: Manuscript “Risk factors for abdominal aortic aneurysm in general populations: A systematic review and meta-analysis”

Dear editor:

“Risk factors for abdominal aortic aneurysm in general populations: A systematic review and meta-analysis”. We would like to thank PLOS ONE for giving us the further opportunity to revise manuscript. We have carefully taken the comments into consideration in preparing our revision, which has resulted in a paper that is clearer and more compelling. The point-by-point responses are attached after this letter. The revisions were highlighted to the text, have been prepared.

The manuscript has not been published previously, in any language, in whole or in part, and is not currently under consideration elsewhere. None of the authors have any competing financial interest to report.

Thank you for considering our manuscript for publication in your esteemed journal.

Point-By-Point Response

Reviewer #1

Introduction:

Question 1: Contextualization of Previous Studies: The introduction provides a general overview of AAA and its risk factors. However, the manuscript would benefit from a more detailed discussion of the gaps in previous studies and the unique contributions of the current study. Please elaborate on the limitations of prior research and explicitly state how this study addresses these gaps.

Response: Thank you for your valuable comments on our manuscript. We fully agree with your suggestion that a more detailed discussion of the limitations of previous studies and the unique contributions of our current study would enhance the quality of the paper. Therefore, we have added this section to our manuscript. Please refer to Para 3, section Introduction, as follows: “Previous studies focused primarily on a few known risk factors and failed to comprehensively cover all potential risk factors. Although Lampsas et al. [11] reported that the relationship between high lipoprotein levels and increased AAA risk is not affected by racial differences, the current study further explored whether there are significant differences in risk factors among different populations, thereby providing a basis for prevention strategies in various regions. The most recent studies that met the inclusion criteria were included in the analysis”

Methods:

Question 1: Search Strategy: While the search strategy is described, the manuscript should include the exact search strings used for each database. This transparency allows for reproducibility of the search process.

Response: Thank you for your valuable feedback. We appreciate your suggestion to enhance the transparency and reproducibility of our search strategy. To address this, we have included the exact search strings used for each database in the manuscript (Supplemental file 1). Here are the details:

Pubmed:

(("Abdominal Aortic Aneurysms"[Mesh]) OR "Aneurysms, Abdominal Aortic"[Mesh] OR "Aortic Aneurysms, Abdominal"[Mesh] OR "Abdominal Aortic Aneurysm"[Title/Abstract] OR "Aneurysm, Abdominal Aortic"[Title/Abstract]) AND ("Screening"[Mesh] OR "Mass Screenings"[Mesh] OR "Screening, Mass"[Title/Abstract] OR "Screenings, Mass"[Title/Abstract] OR "Screenings"[Title/Abstract]) AND ("Factor, Risk"[Mesh] OR "Factors, Risk"[Title/Abstract] OR "Risk Factor"[Title/Abstract] OR "Population at Risk"[Title/Abstract] OR "Risk, Population at"[Title/Abstract] OR "Populations at Risk"[Title/Abstract] OR "Risk, Populations at"[Title/Abstract]) NOT "surgical repair"[Title/Abstract]

EmBase:

('abdominal aortic aneurysm'/exp OR 'aneurysms, abdominal aortic'/exp OR 'aortic aneurysms, abdominal'/exp OR 'abdominal aortic aneurysm':ti,ab OR 'aneurysm, abdominal aortic':ti,ab) AND ('screening'/exp OR 'mass screenings'/exp OR 'screening, mass':ti,ab OR 'screenings, mass':ti,ab OR 'screenings':ti,ab) AND ('risk factor'/exp OR 'factors, risk':ti,ab OR 'risk factor':ti,ab OR 'population at risk':ti,ab OR 'risk, population at':ti,ab OR 'populations at risk':ti,ab OR 'risk, populations at':ti,ab) NOT 'surgical repair':ti,ab

Cochrane library:

(ME "Abdominal Aortic Aneurysms" OR ME "Aneurysms, Abdominal Aortic" OR ME "Aortic Aneurysms, Abdominal" OR "Abdominal Aortic Aneurysm" OR "Aneurysm, Abdominal Aortic") AND (ME "Screening" OR ME "Mass Screenings" OR "Screening, Mass" OR "Screenings, Mass" OR "Screenings") AND (ME "Risk Factors" OR "Factor, Risk" OR "Factors, Risk" OR "Risk Factor" OR "Population at Risk" OR "Risk, Population at" OR "Populations at Risk" OR "Risk, Populations at")

NOT "surgical repair"

Question 2: Study Selection Criteria: The inclusion and exclusion criteria should be more detailed. For example, clarify whether studies without full-text availability were included or excluded, and explain how studies with overlapping populations were handled beyond selecting the most comprehensive one.

Response: Thank you for your valuable feedback. We appreciate your suggestion to provide more detailed inclusion and exclusion criteria. To address your concerns, we have expanded our description of the study selection criteria and provided additional details on how studies with overlapping populations were handled. Please refer to Para 2, section Methods, as follows: “When multiple studies of the same population were published, the most comprehensive study was selected based on several criteria. First, we compared the studies to confirm whether they involved the same or overlapping populations by examining study periods, locations, and participant characteristics. Next, we evaluated the comprehensiveness of each study by considering factors such as robustness of the study design, sample size, duration of follow-up, and completeness and quality of the data. The study with the largest sample size, longest follow-up period, and the most rigorous methodology was selected to ensure the inclusion of the most reliable and extensive data. Reviews and case report articles were removed because of insufficient data for quantitative analysis. Studies that primarily focused on surgical repair techniques without addressing risk factors or screening were excluded”.

Results:

Question 1: Study Characteristics: Provide a more comprehensive table summarizing the characteristics of the included studies (e.g., Outcome investigated, study duration key findings etc.).

Response: Thank you for your valuable feedback. We greatly appreciate your suggestions and have provided a more comprehensive summary of the characteristics of the included studies. To better present this information, we have updated Table 1 to include the study period and the outcomes investigated. Thank you again for your constructive feedback. If you have any further suggestions or need additional information, please let us know.

Question 2: Subgroup Analyses: The manuscript mentions differences in risk factors between Eastern and Western countries but does not provide detailed subgroup analysis results. Include detailed subgroup analysis results and discuss any significant variations found.

Response: Thank you for your valuable feedback. We appreciate your suggestion to include detailed subgroup analysis results and discuss any significant variations found, particularly regarding the differences in risk factors between Eastern and Western countries. While our primary aim was to explore a broad range of risk factors across all included studies, we understand the importance of providing a more detailed subgroup analysis to address the specific differences you mentioned. However, we did not perform a more detailed subgroup analysis in the initial manuscript because our research direction was focused on a comprehensive exploration of all risk factors rather than on a single factor.

Discussion:

Question 1: Implications for Practice and Policy: The manuscript concludes that early intervention and health education are necessary but lacks specific recommendations. Provide more detailed suggestions for clinical practice and public health policies based on the study’s findings.

Response: Thank you for your valuable feedback. We appreciate your suggestion to provide more detailed recommendations for clinical practice and public health policies based on our study's findings. Below, we have included specific suggestions that align with the identified risk factors for AAA. Please refer to Para 2, section Discussion, as follows: “These findings highlight the importance of early intervention and health education for improving modifiable risk factors to reduce the risk of AAA, primarily including: (1) implementing targeted screening programs for high-risk populations, particularly males over 65 years of age, current or former smokers, and individuals with a family history of AAA; (2) establishing regular follow-up protocols to monitor the progression and provide early intervention if necessary for patients identified with small AAAs; (3) risk factor management, including smoking cessation, and hypertension and dyslipidemia management; and (4) health education regarding lifestyle modifications and campaigns to promote awareness of signs and symptoms of AAA”.

Question 2: Limitations: The limitations section should be expanded. Discuss potential biases in study selection, limitations in the quality of included studies, and the potential impact of these limitations on the findings.

Response: Thank you for your valuable feedback. We appreciate your suggestion to expand the limitations section to discuss potential biases in study selection, limitations in the quality of included studies, and the potential impact of these limitations on the findings. Please refer to Para 5, section Discussion, as follows: “This study has several limitations that should be considered when interpreting the findings. First, we included a combination of cross-sectional, retrospective, and prospective studies. This heterogeneity in study design may have introduced selection bias, as different study types have varying levels of methodological rigor and are subject to different types of bias. Moreover, retrospective studies, in particular, are susceptible to recall bias, whereby participants may not accurately remember past exposures or events, potentially leading to misclassification of risk factors. Furthermore, the inclusion of observational studies means that uncontrolled confounding factors may have influenced the results. These confounders could not be fully accounted for in the meta-analysis, potentially leading to biased estimates of the association between risk factors and AAA. Second, most of the included studies followed a cross-sectional design, which limited our ability to establish causality. Moreover, the methodological quality of the included studies varied. Some studies had small sample sizes, limited follow-up periods, and inadequate control of confounding variables, which may have affected the reliability and generalizability of the findings. Third, there was substantial heterogeneity in most of the identified risk factors, which could not be fully explained by sensitivity and subgroup analyses. Finally, as with any meta-analysis, there is the potential for publication bias, whereby studies with significant or positive results are more likely to be published. Moreover, the meta-analyses were based on published articles, which restricted detailed analyses. Further large-scale prospective studies are required to verify the findings of this study and address the limitations mentioned above. Prospective studies with standardized data collection methods and rigorous control of confounding factors would help establish causality and provide more reliable estimates of risk factors for AAA”

Reviewer #2

General comments: In this manuscript, the authors conducted a systematic review and meta-analysis on AAA, which is a common topic in vascular surgery. However, there are many similar systematic reviews and meta-analysis on AAA have already been published, and the majority of risk factors presented in this manuscript have been examined by other analysis, undermining the importance of the current study.

Response: Thank you for your valuable feedback. We appreciate your concern regarding the novelty and importance of our systematic review and meta-analysis on AAA. We understand that there are several existing systematic reviews and meta-analyses on this topic, and many of the risk factors we examined have been previously studied. However, we believe that our study adds significant value to the existing literature for the following reasons: (1) our study includes a more comprehensive and up-to-date analysis, incorporating recent studies that have not been included in previous reviews. This ensures that our findings reflect the most current evidence available, which is crucial given the ongoing advancements in vascular surgery and research; (2) unlike many previous reviews, our study includes a mix of cross-sectional, retrospective, and prospective studies. This diversity in study designs allows for a more robust and comprehensive understanding of the risk factors associated with AAA, as different study types can provide complementary insights; (3) we conducted detailed subgroup and sensitivity analyses to explore the sources of heterogeneity and to assess the robustness of our findings. These analyses provide a deeper understanding of the variations in the data and help to identify specific subgroups where the risk factors may have a more pronounced effect; and (4) we conducted detailed subgroup and sensitivity analyses to explore the sources of heterogeneity and to assess the robustness of our findings. These analyses provide a deeper understanding of the variations in the data and help to identify specific subgroups where the risk factors may have a more pronounced effect.

Question 1: Novelty. What is the novelty of this current research? Most of these have been reported by other papers.

Response: Thank you for your valuable feedback and for raising the question about the novelty of our current research. We understand your concern that many of the risk factors we have examined have been reported in previous studies. However, we believe that our study makes several important and novel contributions to the existing literature on AAA. Our systematic review and meta-analysis include the most recent studies, some of which have not been included in previous reviews. Moreover, we conducted detailed subgroup and sensitivity analyses to explore the sources of heterogeneity and to assess the robustness of our findings. Furthermore, our study incorporates a mix of cross-sectional, retrospective, and prospective studies. This diversity in study designs allows us to provide a more robust and holistic view of the risk factors, as different types of studies can offer unique insights and complement each other's findings.

Question 2: Because most of these have been reported repeatedly, I suggest the authors focus on the limitations of other meta-analysis while doing their own analysis.

Response: Thank you for your valuable feedback and for suggesting that we focus on the limitations of other meta-analyses while conducting our own. We appreciate your insight and agree that addressing the limitations of previous studies can add significant value to our work. Please refer to Para 3, section Introduction, as follows: “Previous studies focused primarily on a few known risk factors and failed to comprehensively cover all potential risk factors. Although Lampsas et al. [11] reported that the relationship between high lipoprotein levels and increased AAA risk is not affected by racial differences, the current study further explored whether there are significant differences in risk factors among different populations, thereby providing a basis for prevention strategies in various regions. The most recent studies that met the inclusion criteria were included in the analysis”

Question 3: There are multiple statements in the manuscript seeming to be more arbitrary than fact based. For example, line237-239:"AAA was more prevalent in males than in female, which could explained by the arterial wall in males is thinner and weaker, making it more susceptible to conditions like atherosclerosis, thus increasing the risk of AAA." what is the evidence for the claim that the arterial wall in males is thinner and weaker?

---

## [Decision Letter · Decision Letter 1]

18 Jul 2025

Risk factors for abdominal aortic aneurysm in general populations: A systematic review and meta-analysis

PONE-D-24-21676R1

Dear Dr. Wu,

We’re pleased to inform you that your manuscript has been judged scientifically suitable for publication and will be formally accepted for publication once it meets all outstanding technical requirements.

Kind regards,

Irena Ilic, MD, PhD

Academic Editor

PLOS ONE

Additional Editor Comments (optional):

Reviewers' comments:

Reviewer's Responses to Questions

**Comments to the Author**

Reviewer #1: All comments have been addressed

2. Is the manuscript technically sound, and do the data support the conclusions?

Reviewer #1: Yes

3. Has the statistical analysis been performed appropriately and rigorously?

Reviewer #1: Yes

4. Have the authors made all data underlying the findings in their manuscript fully available?

Reviewer #1: Yes

5. Is the manuscript presented in an intelligible fashion and written in standard English?

Reviewer #1: Yes

Reviewer #1: Acceptable in the revised format. The authors have submitted a substantially improved version of their manuscript titled “Risk factors for abdominal aortic aneurysm in general populations: A systematic review and meta-analysis.” I have reviewed both the revised manuscript and the detailed point-by-point response to reviewers and find that the authors have addressed nearly all concerns raised in the initial review with care and thoughtfulness.

**Do you want your identity to be public for this peer review?** For information about this choice, including consent withdrawal, please see our Privacy Policy

Reviewer #1: **Yes: ** Pradeep Kumar

---

## [Editor Report · Acceptance letter]

PONE-D-24-21676R1

PLOS ONE

Dear Dr. Wu,

I'm pleased to inform you that your manuscript has been deemed suitable for publication in PLOS ONE. Congratulations! Your manuscript is now being handed over to our production team.

Kind regards,

on behalf of

Dr. Irena Ilic

Academic Editor

PLOS ONE